# Higher-order correlations reveal complex memory in temporal hypergraphs

Luca Gallo [1] ✉, Lucas Lacasa [2], Vito Latora [3,4,5,6] & Federico Battiston [1] ✉

Many real-world complex systems are characterized by interactions in groups that change in time. Current temporal network approaches, however, are unable to describe group dynamics, as they are based on pairwise interactions only. Here, we use time-varying hypergraphs to describe such systems, and we introduce a framework based on higher-order correlations to characterize their temporal organization. The analysis of human interaction data reveals the existence of coherent and interdependent mesoscopic structures, thus capturing aggregation, fragmentation and nucleation processes in social systems. We introduce a model of temporal hypergraphs with non-Markovian group interactions, which reveals complex memory as a fundamental mechanism underlying the emerging pattern in the data.

Temporal networks, where links connecting pairs of nodes are not continuously active, provide a framework to model how the interactions of a complex system evolve in time[1–3]. They have revealed key in understanding how the time-varying interaction network of real-world social and biological systems affects the properties of dynamical processes, such as epidemic spreading[4–7], diffusion[8–12], synchronization[13,14], and others[15–18]. Recent results have highlighted the complex way in which the activity of each link depends on the activities of all other links, showing that memory[19–22] in temporal networks is inherently a multidimensional concept with a well defined microscopic shape[23]. Different approaches have aimed to describe the time evolution of a network as a trajectory in graph space, by naturally extending to the case of graphs notions such as correlations[24] or even dynamical stability[25] traditionally used for scalar or vectorial time-series.

Temporal network approaches, however, have a strong limitation. They are based on a graph description and, as such, they can only describe how dyadic interactions (i.e., links) vary in time, neglecting many-body interactions. Indeed, many real-world social[26–30], biological[31,32], neural[33,34] or ecological[35,36] systems also exhibit higher-order interactions, i.e., interactions involving groups of three or more units at the same time. Such many-body interactions are better modeled by higher-order networks, such as hypergraphs and simplicial complexes, where hyperedges and simplices encode interactions among an arbitrary number of units[37,38]. Interestingly, taking into

account the higher-order architecture of real-world systems is known to produce novel collective phenomena in a variety of dynamical processes, including diffusion[39,40], synchronization[41–45], contagion[46,47] and evolutionary games[48,49].

Some early works have already started to explore the temporal dimension of higher-order interactions. For instance, group interactions in real-world social systems have been found to occur in persistent bursts of activity[28], with events of different sizes close in time also spatially correlated in the network[50]. Such persistent temporal higher-order interactions have been shown to anticipate the onset of endemic states in epidemic processes[51], and to affect the convergence time of nonlinear consensus dynamics[52]. Theoretical frameworks for modeling temporal group activation data[53,54], and for constructing simplicial complexes based on topological data analysis of multivariate time-series from brain functional activity, financial markets and disease spreading[55] have also been recently developed. However, how to analyze and characterize the temporal organization of real-world complex systems with higher-order interactions is, to this day, an open problem.

In this article, we bridge this gap by introducing a general framework to study higher-order temporal dependencies in complex systems. We represent a complex system with interactions in groups whose size and composition can change in time as a temporal hypergraph, i.e., a hypergraph with time-varying hyperedges of different

[1]Department of Network and Data Science, Central European University, Vienna, Austria. [2]Institute for Cross-Disciplinary Physics and Complex Systems (IFISC), CSIC-UIB, Palma de Mallorca, Spain. [3]School of Mathematical Sciences, Queen Mary University of London, London E1 4NS, UK. [4]Department of Physics and Astronomy, University of Catania, 95125 Catania, Italy. [5]INFN Sezione di Catania, Via S. Sofia, 64, 95125 Catania, Italy. [6]Complexity Science Hub Vienna, A-1080 Vienna, Austria. ✉e-mail: gallol@ceu.edu; battistonf@ceu.edu

orders. We then define a set of measures to extract higher-order temporal correlations, namely to characterize how the dynamics of hyperedges of different orders are correlated. We test our framework on a variety of empirical social systems, where patterns are amenable to intuitive interpretations and validation. Results show the existence of long-range correlations at different group sizes and their hierarchical organization. Furthermore, we uncover the presence of temporal correlations between groups of different sizes, i.e., between hyperedges of different orders, unveiling the existence of persistent dynamical relationships between coherent mesoscopic structures previously unaccounted for. Finally, to gain intuition about the underlying microscopic mechanisms, we introduce novel theoretical models of temporal hypergraphs with higher-order memory, able to explain the observed empirical patterns. Beyond networked systems, our measures and models open the door to investigate interactions among emergent coherent structures and other multi-scale phenomena in complex systems.

## Results

### Temporal correlations in hypergraphs

To represent the temporal evolution of systems with higher-order interactions we rely on temporal hypergraphs[28,56]. A temporal hypergraph is a tuple $(\mathcal{V}, \{\mathcal{H}(t)\}_{t=1}^{T})$, where $\mathcal{V}$ is a set of $N$ nodes, and $\{\mathcal{H}(t)\}_{t=1}^{T}$ is a sequence of $T$ sets. Each $\mathcal{H}(t)$ is a set of $M(t)$ hyperedges, representing the interactions among the system units at time $t$. Each hyperedge represents an interaction among multiple units. A hyperedge of order 2, or 2-hyperedge, is a set of two nodes representing a two-body interaction, a 3-hyperedge is a set of three nodes representing a group interaction among three units, and so on, up to order $D$. To study the temporal organization of systems with higher-order interactions, we represent the temporal hypergraphs as a set of $D-1$ sequences $\{\mathbf{A}^{(d)}(t)\}_{t=1}^{T} = \{\mathbf{A}^{(d)}(1), \mathbf{A}^{(d)}(2), \ldots, \mathbf{A}^{(d)}(T)\}$ where the element $a_{ij}^{(d)}(t)$ of matrix $\mathbf{A}^{(d)}(t)$ counts the number of $d$-hyperedges nodes $i$ and $j$ belong to at time $t$, while $a_{ii}^{(d)}(t) = 0 \, \forall i$. See Methods for more details on how to represent temporal hypergraphs.

The presence of higher-order interactions makes the analysis of temporal correlations a multi-faceted problem. First, to quantify temporal correlations in interactions of a given order $d$, we introduce the intra-order correlation matrix

$$\mathcal{C}^{(d)}(\tau) = \frac{1}{T-\tau}\sum_{t=1}^{T-\tau}\frac{1}{(d-1)!^2}\left[\mathbf{A}^{(d)}(t) - \mu^{(d)}\right]\cdot\left[\mathbf{A}^{(d)}(t+\tau) - \mu^{(d)}\right]^{\top}, \quad (1)$$

with $d \in \{2, \ldots, D\}$. Here, $\tau$ is the temporal lag, $\mathbf{A}^{\top}$ denotes the transpose of $\mathbf{A}$, and we have defined the annealed adjacency matrix of order $d$ as $\mu^{(d)} = \frac{1}{T}\sum_{t=1}^{T}\mathbf{A}^{(d)}(t)$. Note that, for $d=2$, Eq. (1) recovers the correlation matrix for temporal networks[24]. The diagonal terms of $\mathcal{C}^{(d)}(\tau)$ capture how hyperedges of order $d$ are temporally autocorrelated, whereas the off-diagonal terms quantify cross-correlations. When the latter are negligible, one can focus on the diagonal terms and define an intra-order correlation function

$$c^{(d)}(\tau) = \text{tr}(\mathcal{C}^{(d)}(\tau)), \quad (2)$$

that provides a scalar measure of how hyperedges of order $d$ are autocorrelated at lag $\tau$.

Second, we can inquire whether interactions of two different orders $d_1$ and $d_2$ display temporal interdependence, i.e., whether mesoscopic structures are dynamically interrelated or, conversely, evolve independently. To this aim, we introduce the cross-order

correlation matrix

$$\mathcal{C}^{(d_1, d_2)}(\tau) = \sum_{t=1}^{T-\tau}\frac{\left[\mathbf{A}^{(d_1)}(t) - \mu^{(d_1)}\right]\cdot\left[\mathbf{A}^{(d_2)}(t+\tau) - \mu^{(d_2)}\right]^{\top}}{(T-\tau)(d_1 - 1)!(d_2 - 1)!}, \quad (3)$$

where $d_1, d_2 \in \{2, \ldots, D\}$. Note that, when $d_1 = d_2 = d$, we recover the intra-order correlation matrix $\mathcal{C}^{(d)}$. We then define a scalar cross-order correlation function as

$$c^{(d_1, d_2)}(\tau) = \text{tr}(\mathcal{C}^{(d_1, d_2)}(\tau)). \quad (4)$$

All the information about intra-order and cross-order correlations can be encoded in a $(D-1) \times (D-1)$ normalized interaction matrix $\mathcal{K}_{d_1 d_2}(\tau) = c^{(d_1, d_2)}(\tau)/2\sqrt{\sigma^{(d_1)}\sigma^{(d_2)}}$, where $\sigma^{(d)} = c^{(d)}(0)$, whose entry $\mathcal{K}_{d_1 d_2}(\tau)$ describes how interactions of order $d_1$ at a given time are correlated with those of order $d_2$ occurring $\tau$ time steps later. Notice that matrix $\mathcal{K}_{d_1 d_2}(\tau)$ is not symmetric, as the quantity $c^{(d_2, d_1)}(\tau)$ measures how order $d_1$ is correlated with order $d_2$ at $\tau$ time steps before, and is in general different from $c^{(d_1, d_2)}(\tau)$. The presence of a significant discrepancy between these two quantities captures asymmetries in the temporal dependencies between different orders of interaction. We quantify such an asymmetry in terms of a cross-order gap function

$$\delta^{(d_1, d_2)}(\tau) = \frac{c^{(d_1, d_2)}(\tau) - c^{(d_2, d_1)}(\tau)}{2\sqrt{\sigma^{(d_1)}\sigma^{(d_2)}}}. \quad (5)$$

A positive value of $\delta^{(d_1, d_2)}(\tau)$ indicates that the presence of groups of size $d_1$ correlates with the presence of groups of size $d_2$ after a time lag $\tau$, more than the other way around.

### Analysis of human interaction data

To explore intra-order and cross-order correlations in complex systems, we consider different social systems, for which we have high-resolution empirical data about their temporal evolution. We first focus on a dataset describing face-to-face interactions over a period of 32h among the $N = 403$ participants of a scientific conference[57,58] (three further cases, namely the social interactions occurring in an office[59], in a hospital ward[60] and in a university campus[61] are described in the Supplementary information). We encode the fine-grained temporal information of the dataset in a temporal hypergraph $(\mathcal{V}, \{\mathcal{H}(t)\}_{t=1}^{T})$, with $|\mathcal{V}| = 403$. Each set $\mathcal{H}(t)$ is constructed by assuming that $d$ individuals in contact at a given time $t$ interact together in a group of size $d$, thus corresponding to a hyperedge of order $d$ at time $t$. See Methods for details on how to reconstruct higher-order interactions from empirical data.

We begin by studying how and if groups of a given size are temporally correlated, i.e., if mesoscopic persistent structures emerge. Figure 1 reports the intra-order correlation functions $c^{(d)}(\tau)$ for orders $d \in \{2, \ldots, 5\}$ (circles). Significant long-range temporal autocorrelations are found for different orders of interaction, as indicated by the slow decays of $c^{(d)}(\tau)$ with $\tau$ in a double logarithmic scale, up to a threshold, which typically decreases with $d$. This indicates that groups, i.e., coherent structures, of larger sizes generally remain autocorrelated for shorter times. Interestingly, we also observe a saturation effect for interactions in groups of size three, with a series of peaks revealing a weak periodicity at large timescales (see Supplementary information for further details). Empirical results are also compared with a null model (squares) obtained by reshuffling the sequence defining the temporal hypergraph.

We then investigate whether interactions in groups of a given size $d_1$ can also be correlated to interactions in groups of size $d_2 \neq d_1$, i.e., whether mesoscopic structures are related to each other. The cross-order correlation functions $c^{(4,5)}(\tau)$ (cyan circles) and $c^{(5,4)}(\tau)$ (olive

circles) for groups of sizes four and five, are reported in Fig. 2a (see Supplementary information for an analysis of other group sizes). For clarity of presentation, we display a binned average of the cross-correlations functions, as well as the corresponding standard deviation. Remarkably, both cross-order correlation functions show precise patterns that can not be reproduced by the corresponding null model (squares), indicating non-trivial relationships between mesoscopic coherent structures. Namely, groups of sizes four and five in this social system show a non-trivial and persistent dependence.

Figure 2b shows the normalized interaction matrix $\mathcal{K}(\tau)$ at time lag $\tau = 600s$ (see Supplementary information for an analysis of different $\tau$). We observe a banded structure around the main diagonal, meaning that cross-order correlations are higher between groups of similar sizes. This indicates that, in the interactions at a scientific conference analyzed here, groups change gradually, with the loss or the addition of one or few members (see Supplementary information for the interaction matrix of different social systems, including the social contacts in a university campus where large groups reveal a more complex correlation pattern). Finally, Fig. 2c shows the cross-order gap function $\delta^{(4,5)}(\tau)$ (purple circles). Positive values of $\delta^{(4,5)}(\tau)$ in almost the entire range of the time lag $\tau$ considered indicate that groups of

size four at a given time are correlated to those of size five occurring $\tau$ time steps later, more than the other way around. This result, which again cannot be reproduced by the null model (green squares), suggests that the formation of a group of five individuals from a group of four is more likely than the loss of one member in groups of five individuals, indicating a preferred temporal direction in the dynamics of group nucleation/fragmentation of this social system (see Supplementary information for an analysis of $\delta^{(d_1,d_2)}(\tau)$ for other group sizes).

The temporal patterns revealed in real-world systems by our framework can be related to other properties of such systems. For instance, the existence of cross-order temporal correlations might explain the presence of overlapping structures, namely the tendency of different hyperedges to share nodes or to be included one within another, observed in temporally-aggregated hypergraphs[26,29,62,63].

## Models of hypergraphs with higher-order memory

To investigate the mechanisms shaping intra-order and cross-order correlation profiles, we introduce two models to generate synthetic temporal hypergraphs with higher-order memory, inspired by DAR processes[21,23,64]. The first model, named Discrete Auto Regressive Hypergraph (DARH) model, treats the binary states of each hyperedge $h^\alpha \in \{0, 1\}$ (absent/present) as independent stochastic processes. Each hyperedge updates its state either drawing a state from its past, or randomly sampling a new state. With probability $q^{(d)}$, where $d \in \{2, ..., D\}$, a hyperedge of order $d$ samples its state uniformly at random from its $m_s^{(d)}$ previous states, while with probability $1 - q^{(d)}$, the hyperedge state is drawn randomly following a Bernoulli process with probability $y^{(d)}$. In this way, the tuning parameter $q^{(d)}$ controls the memory strength[21] of the hyperedges of order $d$. See Methods for a detailed description of the DARH model. Our first model displays intra-order correlations (i.e., the existence of mesoscopic persistent structures), but no cross-order correlations (i.e., these coherent structures do not interact). See Supplementary information for a characterization of the DARH model.

We then introduce a second model, the cross-memory DARH (cDARH) model, a variation of the DARH model where a hyperedge of order $d$ can update its state by drawing not only from its $m_s^{(d)}$ previous states but also from the $m_c^{(d',d)}$ previous states of a hyperedge of a different order $d'$. This updating mechanism, to which we will refer as cross-order memory, is what ultimately allows the model to account for interactions among coherent structures. The parameter $m_s^{(d)}$ represents the intra-order memory length of the hyperedges of order $d$, while $m_c^{(d',d)}$ is the cross-order memory length. When copying from memory, each hyperedge draws from the past of other hyperedges with probability $p^{(d)}$, and from its own past with probability $1 - p^{(d)}$. We assume that hyperedges can copy from the memory of overlapping

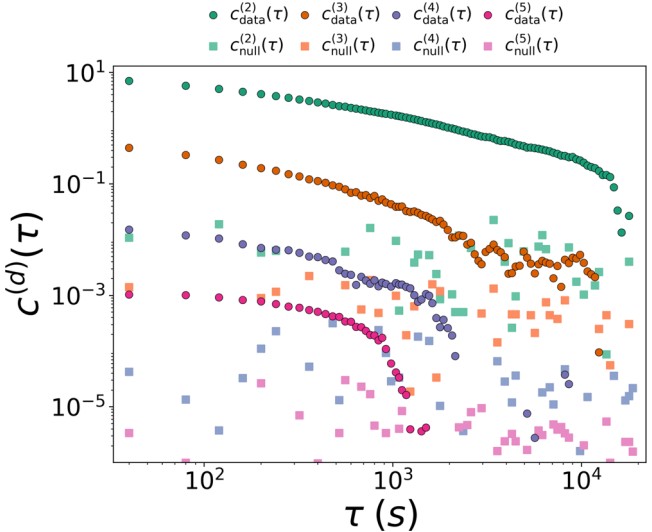

**Fig. 1 | Intra-order correlations in human face-to-face interactions.** Circles show the value of the intra-order correlation functions $c^{(d)}(\tau)$ for interactions in groups of size $d \in \{2, ..., 5\}$. Squares refer to a randomized null model where temporal correlations have been removed by reshuffling time steps.

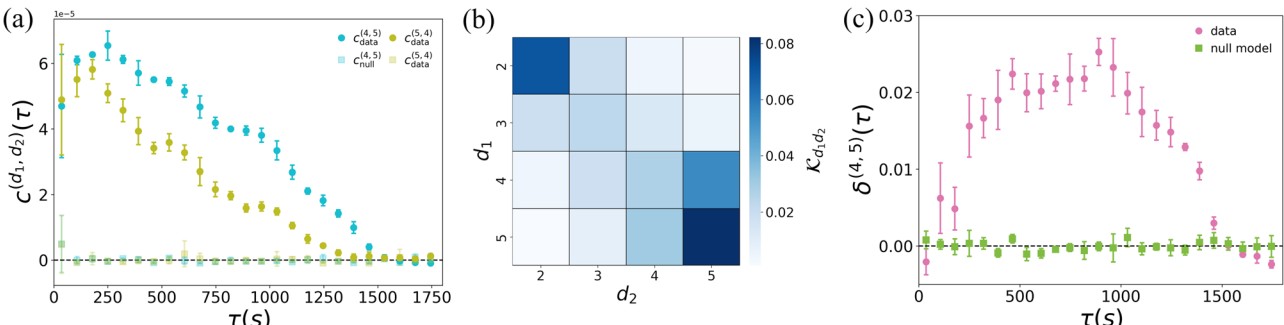

**Fig. 2 | Cross-order correlations in human face-to-face interactions. a** Cross-order correlation functions, $c^{(4,5)}(\tau)$ (cyan) and $c^{(5,4)}(\tau)$ (olive), describing the temporal dependencies between interactions of order four and interactions of order five. We compare the empirical system (circles) with a randomized null model with reshuffled time-steps (squares). **b** Normalized interaction matrix $\mathcal{K}_{d_1 d_2}(\tau)$, encoding the temporal dependencies between any pairs of order $d_1, d_2 \in \{2, ..., 5\}$ at time lag $\tau = 600s$. **c** Cross-order gap function $\delta^{(4,5)}(\tau)$ (purple circles), compared with the null model (green squares). The values of the cross-order correlation and the cross-order gap functions are binned averaged, with the error bars representing the standard deviation.

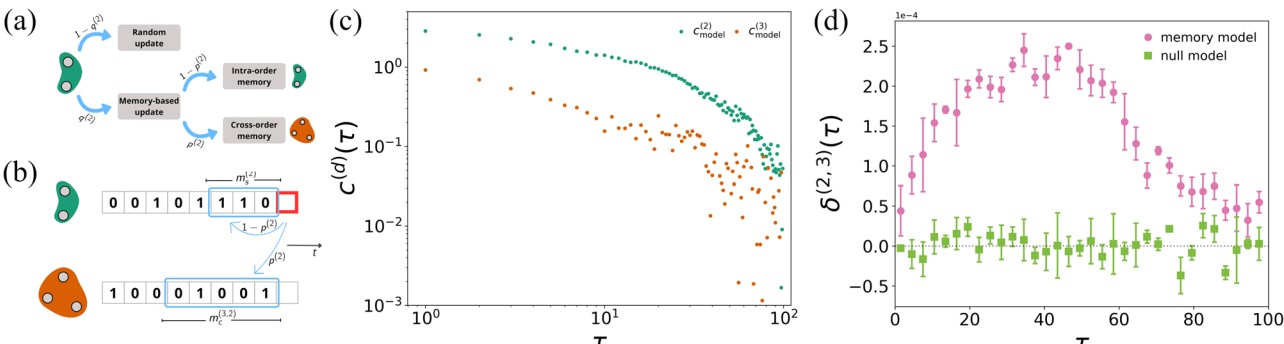

**Fig. 3 | Temporal correlations in the cross-memory Discrete Auto Regressive Hypergraph (cDARH) model. a** An illustration of the cDARH model. At each time $t$, the state of a hyperedge of order $d$ is updated either randomly, with a probability $1 - q^{(d)}$, or through a memory-based process, with a probability $q^{(d)}$ ($d = 2$ in the illustration). In the latter case, the state is updated by copying either a previous state from the past of the hyperedge (intra-order memory) or a previous state of an overlapping hyperedge of a different order (cross-order memory), according to a probability $p^{(d)}$. **b** A schematic illustration of the memory-based state update of a 2-hyperedge in the cDARH model. With probability $1 - p^{(2)}$, the state is drawn from

one of the $m_s^{(2)}$ previous states of the hyperedge, while with probability $p^{(2)}$ it is sampled from one of the $m_c^{(3,2)}$ previous states of a overlapping hyperedge of order 3. **c** Intra-order correlations $c^{(d)}$, with $d \in \{2, 3\}$, for hyperedges of order two (green circles and squares) and three (orange circles and squares). **d** Cross-order gap function $\delta^{(2,3)}$ between hyperedges of order two and three. We compare intra-order and cross-order correlations for the hypergraph generated using the cDARH model with a randomized null model. The values of the cross-order gap functions are binned averaged, with the error bars representing the standard deviation.

hyperedges only. This choice is motivated by previous studies on higher-order interactions in social networks pointing out a tendency of groups to progressively add or remove members, one step at a time[28]. For illustration, let us consider the case of groups of size two and three: a 2-hyperedge $\{i, j\}$ selects one of the $(N - 2)$ possible 3-hyperedges containing nodes $i$ and $j$ and draws from its previous $m_c^{(3,2)}$ states. Similarly, a 3-hyperedge $\{i, j, k\}$ selects one of the three 2-hyperedges that can be formed from it, i.e., $\{i, j\}$, $\{j, k\}$, and $\{i, k\}$, and copies a state from its previous $m_c^{(2,3)}$ steps (see Fig. 3a–b for a schematic illustration of the model). Such a mechanism can be straightforwardly extended to hyperedges of other orders. See Methods for more details about the cDARH model.

We generate with the cDARH model temporal hypergraphs with $N = 10$ nodes, maximum hyperedge order $D = 3$ and a temporal range of $T = 3 \cdot 10^4$ time steps. Note that real-world systems are usually characterized by a larger number of units. However, generating a temporal hypergraph with a realistic size using the cDARH model can be computationally costly. Yet our model is able to describe the patterns observed in the data even considering a few nodes. See Supplementary information for an analysis of larger hypergraphs. We set $p^{(2)} = 0$ and $p^{(3)} = 0.6$, meaning that hyperedges of order three can copy from the past of hyperedges of order two, while hyperedges of order two evolve independently. We also set $m_c^{(2,3)} = 60$, for the cross-order memory length of 3-hyperedges. Considering a single couple of intra-order memory lengths $m_s^{(2)}, m_s^{(3)}$ for all hyperedges of order two and three, respectively, the cDARH model displays both intra-order and cross-order correlations. Yet the profiles of $c^{(d)}(\tau)$ do not match exactly those of empirical data. In particular, the functions $c^{(2)}$ and $c^{(3)}$ remain constant for $\tau \le m_s^{(d)}$ and decay exponentially after that value, with the same rate of decay (see Supplementary information for a deeper analysis), while in empirical systems the correlation functions follow a power-law decay. This is not surprising, as real-world social interactions can be shaped by different scales of memory[23]. Hence, we sample the intra-order memory length of a hyperedge of order $d$ from a uniform distribution, with maximum values for the interactions of orders two and three set to $m_{s,\max}^{(2)} = 40$, and $m_{s,\max}^{(3)} = 10$, respectively. In Fig. 3c we observe that the profile of the intra-order correlation functions $c^{(d)}$ shows a slow decay followed by a loss of correlation, which is in good agreement with what we see in the empirical networks. Such a minimal model reveals that memory can be the driving mechanism for the emergence of intra-order temporal correlations, with different orders possessing different degrees of memory, also explaining the

hierarchical structure of correlation observed in the data. Figure 3d shows that $\delta^{(2,3)}(\tau) > 0$ for different values of $\tau$ (purple circles), meaning that hyperedges of order two are correlated to hyperedges of order three occurring later in time more than the other way around. We observe a striking similarity between this trend and that observed in Fig. 2c for the empirical data. This result indicates that cross-order memory is a fundamental factor for the emergence of cross-order correlations among different orders of interaction as well as cross-order gaps in real-world social systems. The two peaks for $\delta^{(4,5)}(\tau)$ observed in the empirical system suggest again a more complex dependence on memory, possibly due to multiple temporal scales.

## Discussion

In this article, we have introduced a framework to characterize different dimensions of memory in networked systems with higher-order interactions. We have shown that real-world social systems display long-range temporal correlations at different group sizes –i.e., that coherent mesoscopic structures emerge–, organized in a hierarchy across multiple scales. Moreover, we have found that group interactions are characterized by non-trivial cross-order correlations, with cross-memory being a fundamental mechanism underlying such a complex behavior. In the context of social systems, such cross-order interactions can be interpreted in terms of the schisming phenomenon[65–68], where groups in human interactions, e.g. conversations, fluctuate, nucleate, and display complex dynamics.

In conclusion, our work sheds light on the multifaceted nature of memory that emerges at different scales in real-world interacting systems. The analyses presented here can be naturally extended to other higher-order complex systems traditionally modeled in terms of networks of interactions, such as the human brain and biological ecosystems. Beyond the scope of network science, we hope that our framework can open new avenues to reveal the higher-order dynamics of coherent structures in a variety of physical systems, from multi-fragmentation in nuclear physics to vortex-vortex interaction in the atmosphere or other fluid dynamical systems.

## Methods
### Representation of time-varying systems with higher-order interactions
Systems with higher-order interactions can be represented as hypergraphs[56]. A hypergraph is a tuple $(\mathcal{V}, \mathcal{H})$, where $\mathcal{V}$ is a set of $N$ nodes, and $\mathcal{H}$ is a set of $M$ hyperedges. Each hyperedge represents an

interaction among two or more units. A hyperedge of order 2, or 2-hyperedge, is a set of two nodes representing a two-body interaction, a 3-hyperedge is a set of three nodes representing an interaction among three units, and so on, up to order $D$. While hypergraphs are usually represented by adjacency tensors of different ranks, to capture dynamical dependencies within and among orders we will rely instead on a set of adjacency matrices of the same rank. First, we consider a set of incidence matrices $[\mathbf{E}^{(2)}, \mathbf{E}^{(3)}, ..., \mathbf{E}^{(D)}]$ where the element $e_{i\alpha}^{(d)}$ of matrix $\mathbf{E}^{(d)}$ is one if node $i$ belongs to the $d$-hyperedge $\alpha$, while it is zero otherwise. For each order of interaction $d$, we can then construct an adjacency matrix $\mathbf{A}^{(d)}$ as

$$\mathbf{A}^{(d)} = \mathbf{E}^{(d)}\mathbf{E}^{(d)\top} - \mathrm{diag}(\mathbf{E}^{(d)}\mathbf{E}^{(d)\top}). \qquad (6)$$

The off-diagonal elements $a_{ij}^{(d)} = \sum_\alpha e_{i\alpha}e_{j\alpha}$ represents the number of $d$-hyperedges nodes $i$ and $j$ belong to, while $a_{ii}^{(d)} = 0 \,\forall i$.

To represent a system with higher-order interactions evolving in time we rely on temporal hypergraphs[28]. A temporal hypergraph is a tuple $(\mathcal{V}, \{\mathcal{H}(t)\}_{t=1}^T)$, where $\mathcal{V}$ is again a set of $N$ nodes, and $\{\mathcal{H}(t)\}_{t=1}^T$ is a sequence of $T$ sets. Each $\mathcal{H}(t)$ is a set of $M(t)$ hyperedges, representing the interactions occurring at time $t$. For each order of interaction $d$, we can define a sequence $\{\mathbf{A}^{(d)}(t)\}_{t=1}^T = \{\mathbf{A}^{(d)}(1), \mathbf{A}^{(d)}(2), ..., \mathbf{A}^{(d)}(T)\}$, where $\mathbf{A}^{(d)}(t)$ is an adjacency matrix encoding the interactions of order $d$ occurring at time $t$. Hence, we can fully represent the temporal evolution of the system using a set of $D-1$ sequences $[\{\mathbf{A}^{(2)}(t)\}_{t=1}^T, \{\mathbf{A}^{(3)}(t)\}_{t=1}^T, ..., \{\mathbf{A}^{(D)}(t)\}_{t=1}^T]$.

## Reconstruction of higher-order social interactions from empirical data

To investigate the temporal organization of social interactions, we rely on four datasets, three coming from the SocioPatterns project[57–60] and one from the Copenhagen Network Study[61]. These datasets store the interactions among the individuals as a temporal network, namely they contain dyadic interactions only. However, as people often engage in groups where more than two individuals interact at the same time, a network description of the system might result in an inadequate representation of the system. Still, the fine-grained temporal information of the datasets allows us to extract group interactions from the data. In particular, we assume that $d$ individuals that are in contact through dyadic interactions at a given time $t$ interact together in a group of size $d$. For instance, if at time $t$ individual $i$ is in contact with individuals $j$ and $k$, while individual $j$ is also interacting with individual $k$, we assume individuals $i, j$ and $k$ to be engaged in a group interaction. Mathematically, if at time $t$ a set of $d$ nodes form a clique in the temporal network, we promote the clique to a $d$-hyperedge in the temporal hypergraph.

## The DARH and the cDARH models

To investigate the mechanisms shaping the onset of intra-order and cross-order correlations in temporal hypergraphs, we introduce two theoretical models that generate temporal hypergraphs with higher-order memory. The first model, called the Discrete Auto Regressive Hypergraph (DARH) model, treats the binary states of each hyperedge $h^\alpha \in \{0, 1\}$ (absent/present) as independent stochastic processes. The state of each hyperedge is updated either randomly or by drawing one of the previous states of the hyperedge. In particular, with a probability $1-q^{(d)}$, the hyperedge state is drawn randomly according to a Bernoulli process, i.e., the hyperedge is present with a probability $y^{(d)}$, or it is absent with a probability $1-y^{(d)}$. Note that $d \in \{2, ..., D\}$ represents the order of the hyperedge, meaning that for each order of interaction separately we can tune the sampling probabilities $q^{(d)}$ and $y^{(d)}$. With probability $q^{(d)}$, instead, the next state is sampled uniformly at random from the $m_s^{(d)}$ previous states of the hyperedge. Formally, the dynamics

of the state of a hyperedge $\alpha$ of order $d$, $h^\alpha$, is given by

$$h_t^\alpha = Q_t h_{t-\mu_t}^\alpha + (1-Q_t)Y_t \qquad (7)$$

where $Q_t \sim \text{Bernoulli}(q^{(d)})$ is a random variable selecting how the state of the hyperedge is updated, $Y_t \sim \text{Bernoulli}(y^{(d)})$ defines whether the hyperedge is present/absent when its state is selected randomly, while $\mu_t \sim \text{Uniform}(1, m_s^{(d)})$ determines which state is drawn when the update is done by sampling from the hyperedge past.

The second model, named the cross-memory DARH (cDARH) model, is a variation of the DARH model where a hyperedge of order $d$ can update its state by drawing not only from its past but also from that of a hyperedge of a different order. Similar to the DARH model, with a probability $1-q^{(d)}$, the hyperedge state is drawn randomly according to a Bernoulli process with probability $y^{(d)}$, while the state is copied from past states with a probability $q^{(d)}$. When copying from memory, with probability $1-p^{(d)}$ the state of the hyperedge is sampled uniformly at random from its $m_s^{(d)}$ previous states. With probability $p^{(d)}$, instead, the state of the hyperedge is drawn from the $m_c^{(d',d)}$ previous states of an overlapping hyperedge of order $d'$ that overlaps. The order $d'$ of the hyperedge can be drawn according to a given probability distribution $\rho^{(d)}(d')$. Formally, we can write the dynamics of the state of a hyperedge $\alpha$ of order $d$, $h^\alpha$, as

$$h_t^\alpha = Q_t h_{t-\mu_t}^{\varepsilon_t(\alpha)} + (1-Q_t)Y_t. \qquad (8)$$

As for the DARH model, $Q_t \sim \text{Bernoulli}(q^{(d)})$ is a random variable selecting how to update the state of the hyperedge, while $Y_t \sim \text{Bernoulli}(y^{(d)})$ determines if the hyperedge is present/absent when sampling randomly. $\varepsilon_t(\alpha)$ is a random variable that defines if the update of the hyperedge is done by sampling from its own past or from that of another hyperedge. Mathematically, it follows the equation

$$\varepsilon_t(\alpha) = P_t\beta + (1-P_t)\alpha, \qquad (9)$$

where $P_t \sim \text{Bernoulli}(p^{(d)})$ selects whether to copy from the past of hyperedge $\alpha$ or from that of another hyperedge, indexed as $\beta$. $\beta$ is sampled from the set of hyperedges of order $d'$ overlapping with $\alpha$, with $d'$ drawn from $\rho^{(d)}(d')$. Finally, the variable $\mu$, determining which state from the past is sampled when copying from memory, is sampled according to the value of $\varepsilon_t(\alpha)$. In particular, when $\varepsilon_t(\alpha) = \alpha$, i.e., for the intra-order memory process, we have $\mu_t \sim \text{Uniform}(1, m_s^{(d)})$, while we have $\mu_t \sim \text{Uniform}(1, m_c^{(d',d)})$ when $\varepsilon_t(\alpha) \neq \alpha$, i.e., for the cross-order memory process.

## Reporting summary

Further information on research design is available in the Nature Portfolio Reporting Summary linked to this article.

# Data availability

The SocioPatterns data on the contacts in the scientific conference, the office and the hospital ward are available at https://www.sociopatterns.org/datasets. The Copenhagen Network Study data on the contacts in the university campus are available at https://doi.org/10.6084/m9.figshare.11283407.

# Code availability

The measures described here are implemented as part of the HGX library[69] and are available at https://github.com/HGX-Team/hypergraphx.

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

## Acknowledgements

L.G. acknowledges C. Zappalà for the insightful discussions and comments. L.G. and F.B. acknowledge support from the Air Force Office of Scientific Research under award number FA8655-22-1-7025. L.L. acknowledges funding from the Spanish Research Agency MCIN/AEI/ 10.13039/501100011033 via projects DYNDEEP (EUR2021-122007), MISLAND (PID2020-114324GB-C22), and the María de Maeztu project CEX2021-001164-M. V.L. acknowledges support from the European Union, NextGenerationEU, GRINS project (GRINS PE00000018 - CUP E63C22002120006).

## Author contributions

L.G. conceptualized the work, developed the methodology, carried out the analysis, curated the data, the code, and the visualization. L.L. provided methodological insights and carried out the formal analysis. V.L. provided methodological insights. F.B. conceptualized and supervised the work. All authors wrote, reviewed and edited the paper.

## Competing interests

The authors declare no competing interests.
