## [Peer Review File · Nature Communications]

REVIEWER COMMENTS

Reviewer #1 (Remarks to the Author):

Gallo and colleagues present here a novel framework to study temporal dynamics and memory in hypergraphs, apply it to empirical social networks, and present a streamlined modeling framework to account for such effects. This is a highly timely topic, with potential for large impact.

The manuscript presents this complex topic in a clear manner, suitable for a general audience, and it is very nicely written. Whilst the analyses included in the study are somewhat simple, they serve to illustrate in a clear way the methods presented, which I think will inspire. Overall, I think this paper has the potential to inspire further research, developing this framework further and applying it to systems beyond social networks. Thus, I strongly recommend its publication in Nature Communications.

I point out a few minor points to improve the manuscript, I do not need to see a revised version:

- I have spotted a few typos:

- In the abstract, I think there is some grammar error in the last sentence “as a fundamental mechanism underlying the pattern in the data.” I suggest something like “as a fundamental mechanism underlying the emerging patterns in the data”, or something along that line.

- At the end of the introduction, in the sentence “unveiling the existence of persistent dynamical interactions ... previously unaccounted.”, it should be “unaccounted for”.

- In the first paragraph of the results, “up to a order D ” should be “up to order D ”.

- In the results section “Analysis of human interactions data”, in the paragraph, there is a space after the parenthesis in “(three further cases,...”. I also recommend removing the “s” in “interactions” in the section title.

- There are several mentions to “interactions” between different orders in the manuscript. I am not sure if this is the appropriate term for the phenomenon under study, which corresponds really with the evolution of group sizes in small (typically one) increments, as discussed by the authors in the manuscript. To me, interactions would imply the merge or division of groups, that is, an interaction between groups of size 5 and 4 would be the merge of these group creating a group of size 9, whereas in the manuscript it refers to the decay of a group of size 5 into a group of size 4, following the leave of one member. The authors are free to keep interaction if they choose to, but I would then define what they mean by it already in the introduction, since at the moment it is only clearly stated in the results section.

Reviewer #3 (Remarks to the Author):

This is a solid contribution on how to capture the temporal properties of multiway networks. The authors extend measures and models designed for temporal networks to the case when interactions involve more than two nodes, and then convincingly show on empirical data the novel insight provided by the framework. The manuscript is overall well written and the numerical experiments support the theoretical findings. This is, in my view, a promising approach for the mining of relational data and I am happy to recommend it for publication.

As a minor comment, I would recommend the authors to discuss relations of their work to previous approaches. In particular, on how to capture the relations between different hyper edges. In this manuscript, the authors consider a cross-order correlation, but does this relate to previous methods where the authors investigated - on static hypergraphs - the tendency for hyper-edges to share nodes? If not, how could this notion be included in the setting proposed in this submission?

On this topic, I can think of the recent papers:

Landry N W, et al. 2023 The simpliciality of higher-order networks (arXiv:2308.13918)

Larock, T et al 2023 Encapsulation structure and dynamics in hypergraphs, Journal of Physics: Complexity

Which relate to the following paper on the notion of closure (for higher-order networks) cited by the authors:

Benson A R, et al 2018 Simplicial closure and higher-order link prediction Proc. Natl Acad. Sci. 115 E11221–30

Dear Referees,

thank you for your careful reading of our manuscript, your constructive comments, and for recommending publication of our work in Nature Communications. In the following, we provide a detailed point-by-point response to your comments. All edits are reported in orange in the new manuscript files.

The authors

Response to Reviewer 1

Gallo and colleagues present here a novel framework to study temporal dynamics and memory in hypergraphs, apply it to empirical social networks, and present a streamlined modeling framework to account for such effects. This is a highly timely topic, with potential for large impact. The manuscript presents this complex topic in a clear manner, suitable for a general audience, and it is very nicely written. Whilst the analyses included in the study are somewhat simple, they serve to illustrate in a clear way the methods presented, which I think will inspire. Overall, I think this paper has the potential to inspire further research, developing this framework further and applying it to systems beyond social networks. Thus, I strongly recommend its publication in Nature Communications.

I point out a few minor points to improve the manuscript, I do not need to see a revised version:

- I have spotted a few typos:
 - In the abstract, I think there is some grammar error in the last sentence “as a fundamental mechanism underlying the pattern in the data.” I suggest something like “as a fundamental mechanism underlying the emerging patterns in the data”, or something along that line.
 - At the end of the introduction, in the sentence “unveiling the existence of persistent dynamical interactions . . . previously unaccounted.”, it should be “unaccounted for”.
 - In the first paragraph of the results, “up to a order D” should be “up to order D”.
 - In the results section “Analysis of human interactions data”, in the paragraph, there is an space after the parenthesis in “(three further cases, . . .”. I also recommend removing the “s” in “interactions” in the section title.
 - There are several mentions to “interactions” between different orders in the manuscript. I am not sure if this is the appropriate term for the phenomenon under study, which corresponds really with the evolution of group sizes in small (typically one) increments, as discussed by the authors in the manuscript. To me, interactions would imply the merge or division of groups, that is, an interaction between groups of size 5 and 4 would be the merge of these group creating a group of size 9, whereas in the manuscript it refers to the decay of a group of size 5 into a group of size 4, following the leave of one member. The authors are free to keep interaction if they choose to, but I would then define what they mean by it already in the introduction, since at the moment it is only clearly stated in the results section.

A: We thank the Reviewer for carefully assessing our manuscript and for the suggestions that helped us improve the presentations of our results. We have corrected all the mentioned typos and checked the correctness and clarity of our writing throughout the manuscript. We have also followed the suggestion not to use the term “interaction” when referring about the (complex) relation among the different orders, so to improve the clarity of the text.

We remark that the revised version of the manuscript reports a figure that was previously contained in the Supplemental Material. In particular, the new Fig. 3 now reports the intra-order and cross-order temporal correlations of a hypergraph generated with the cDARH model where the intra-order memory parameter is sampled from a uniform distribution. The choice for this edit is that we have decided to present a more realistic model, able to better describe the patterns observed in the data (in

particular, the slow-decay of intra-order correlation functions), in the manuscript. The previous figure, which showed the result obtained for the simpler version of the cDARH model, where a single memory parameter is used for each order of interaction, has been moved to the SM. The text of the manuscript has been changed accordingly.

Response to Reviewer 3

This is a solid contribution on how to capture the temporal properties of multiway networks. The authors extend measures and models designed for temporal networks to the case when interactions involve more than two nodes, and then convincingly show on empirical data the novel insight provided by the framework. The manuscript is overall well written and the numerical experiments support the theoretical findings. This is, in my view, a promising approach for the mining of relational data and I am happy to recommend it for publication.

As a minor comment, I would recommend the authors to discuss relations of their work to previous approaches. In particular, on how to capture the relations between different hyper edges. In this manuscript, the authors consider a cross-order correlation, but does this relate to previous methods where the authors investigated - on static hypergraphs - the tendency for hyper-edges to share nodes? If not, how could this notion be included in the setting proposed in this submission?

On this topic, I can think of the recent papers:

Landry N W, et al. 2023 The simpliciality of higher-order networks (arXiv:2308.13918)

Larock, T et al 2023 Encapsulation structure and dynamics in hypergraphs, Journal of Physics: Complexity

Which relate to the following paper on the notion of closure (for higher-order networks) cited by he authors:

Benson A R, et al 2018 Simplicial closure and higher-order link prediction Proc. Natl Acad. Sci. 115 E11221–30

A: We thank the Reviewer for carefully reading our manuscript and for the relevant comment. We agree with the Reviewer that the presence of significant temporal correlations in the interactions is likely to be associated with a high tendency of hyperedges to share nodes when considering a temporally-aggregated hypergraph. As also pointed out by the Reviewer, previous works have indeed highlighted that some of the social systems that we have analyzed in our work are characterized by nested and encapsulated structures. For instance, Landry N.W. et al. showed that contact networks have high levels of simpliciality, as shown in Fig. 1.

Figure 1: The simpliciality of empirical datasets. Hypergraph constructed from face-to-face interaction data tend to have high level of simpliciality. Figure retrieved from Landry N.W. et al., arXiv:2308.13918 (2023).

In particular, we remark that *hospital-lyon* corresponds to the hospital dataset we have discussed in our paper.

Similarly, Lotito Q.F. et al. showed that social systems shows an over-expression of higher-order motifs displaying nested structure among hyperedges of different orders, a concept that can be related to the presence of structures with a high-level of encapsulation/simplicity. Fig. 2 shows the results relative to higher-order motifs involving three nodes.

Figure 2: Significance profiles (SP) of hypergraphs from higher-order motifs of order 3. Δ represents the abundance of each motif relative to random networks. Figure retrieved from Lotito Q.F. et al., Communications Physics 5, 79 (2022).

Here, the *conference*, *workplace*, and *hospital* systems correspond to the same datasets we used in our work. Note that similar results apply for motifs involving four nodes, with the more nested structures over-represented compared to a random scenario.

Finally, La Rock T. et al. revealed the presence of high level of encapsulation in other contact networks (face-to-face interactions in a primary school and a high-school).

The connection between the presence of temporal correlations and the nestedness of the hyperedges may lay in the microscopic dynamics of conversational groups. As we have discussed in our paper, the existence of significant cross-order temporal correlations between similar orders of interactions is associated to the the gradual gain (or loss) of group members. Consequently, in the temporally-aggregated hypergraph, the hyperedges representing the groups before and after these kind of transitions will be encapsulated one into the other. Of course, other group dynamics can be at play in these systems, thus making the relationship between temporal correlations and hyperedge overlapping potentially more complex. In summary, while social contact data display both cross-temporal correlations and highly nested pattern when data are temporally aggregated, the causal relationship between these two properties still has to be clarified, and deserves further investigation..

Overall, the point raised by the Reviewer is of great interest, and to take this comment into account

we have added a paragraph in the revised version of the manuscript discussing the topic.

We remark that the revised version of the manuscript reports a figure that was previously contained in the Supplemental Material. In particular, the new Fig. 3 now reports the intra-order and cross-order temporal correlations of a hypergraph generated with the cDARH model where the intra-order memory parameter is sampled from a uniform distribution. The choice for this edit is that we have decided to present a more realistic model, able to better describe the patterns observed in the data (in particular, the slow-decay of intra-order correlation functions), in the manuscript. The previous figure, which showed the result obtained for the simpler version of the cDARH model, where a single memory parameter is used for each order of interaction, has been moved to the SM. The text of the manuscript has been changed accordingly.

REVIEWERS' COMMENTS

Reviewer #3 (Remarks to the Author):

The authors have successfully addressed my minor comments. I am happy to recommend their manuscript for publication.